# Impact of the COVID-19 Pandemic on Estonian Elite Athletes: Survey on Mental Health Characteristics, Training Conditions, Competition Possibilities, and Perception of Supportiveness

**DOI:** 10.3390/ijerph18084317

**Published:** 2021-04-19

**Authors:** Ülle Parm, Anu Aluoja, Tuuli Tomingas, Anna-Liisa Tamm

**Affiliations:** 1Physiotherapy and Environmental Health Department, Tartu Health Care College, 50411 Tartu, Estonia; nooruse@nooruse.ee (Ü.P.); tuuli.tomingas@gmail.com (T.T.); 2Department of Psychiatry, Faculty of Medicine, University of Tartu, 50417 Tartu, Estonia; anu.aluoja@ut.ee

**Keywords:** elite athletes, COVID-19 pandemic, anxiety, depression

## Abstract

*Background*: The postponement and cancellation of the competition season due to COVID-19 could cause significant mental health problems for an elite athlete. The aim of this study was to describe the mental health characteristics of Estonian elite athletes, their training conditions, competition possibilities, and the support they received during COVID-19. *Methods*: Athletes completed self-reported questionnaires (including Emotional State Questionnaire). The authors applied descriptive statistics, t-test, and χ^2^ test for comparison of study groups (*p* < 0.05). *Results*: Altogether 102 athletes (♂ = 44) were surveyed. The most disturbing issue for athletes was the closing of training centers (57.8%) and cancellation of competitions (50%); 64.7% of athletes reported a negative response from not being able to visit healthcare specialists. Fortunately, athletes could receive virtual coaching. Two-thirds of the athletes had some indication of distress (♀ > ♂): 25% of males and 39.7% of females had symptoms indicating depression; ♀ = 27.6%, ♂ = 13.6% anxiety; ♀ = 56.9%, ♂ = 31.8% fatigue (*p* = 0.021); ♀ = 55.1%, ♂ = 27.2% insomnia (*p* = 0.009); 27.5% thought about ending their career (frequency in high distress group compared with low: *p* = 0.022); and 2.9% were certain they would stop their training completely. Family members were the biggest emotional supporters; 16.7% did not get support from anyone. *Conclusion*: The Estonian sport community needs to adapt to life in a pandemic environment and help athletes to maintain training and competition activities and in turn, their mental health.

## 1. Introduction

By March 2021, more than 113,500,000 cases of SARS-CoV-2-virus (COVID-19) were confirmed globally, including more than 2,500,000 deaths [1]. In the beginning of June 2020, when the authors conducted this study, 6,414,476 cases of COVID-19 were reported in 188 countries [2]. At the same time, the situation in Estonia was rather moderate, with approximately 10 new cases daily, but the pandemic was emerging, people were terrified, and the entire community was under lockdown with nearly all training centers closed. The COVID-19 outbreak, fear of infection and prolonged quarantine could lead to health problems such as stress, anxiety, depressive symptoms, insomnia, denial, anger, fear [3], stigmatization, low self-esteem [4], and also long-lasting post-traumatic stress globally [5].

Competitive sport during the pandemic situation was tremendously affected. International and national (including local) sports events were canceled or postponed to minimize the risk of viral spread. Prior to the pandemic, elite athletes had been carefully preparing for international competitions (including the Olympic and Paralympic Games, Tokyo). For some athletes, the opportunity to compete in the Olympics was gone and would never arise again [6]. According to Toresdahl and Asif [7], the season’s suspension and competition cancellation could cause significant grief, stress, anxiety, frustration, and sadness for an athlete.

Training opportunities were not the same as under normal circumstances. Some athletes trained at home on their own and mostly unsupervised [8]; some athletes trained in training camps far from home under mandatory isolation along with excessive unoccupied time and no clear work target [6]. Athletes were likely exposed to some level of detraining and inappropriate training stimuli [8].

According to Håkansson et al. [9], COVID-19-related distress associated with mental health symptoms is common among elite athletes, so ensuring the mental health and wellbeing of elite athletes is important [6]. Continuing with training is an important component for mental health protection of the athlete, particularly to reduce the risk of anxiety and depression [6]. The aim of the study was to describe the mental health characteristics (depression, anxiety, fatigue, and insomnia) of the athletes, identify the types of support they received, and training conditions and competition during the COVID-19 pandemic.

## 2. Materials and Methods

The Committee of Ethics of the University of Tartu approved the study (protocol no. 313/T-2, 30 April–4 May 2020). The questionnaire (via electronic system connect.ee) was sent to all Estonian elite athletes (approximately 600) by the contact person of the Estonian Olympic Committee. Elite Estonian athletes filled in the electronic questionnaire from May to June 2020. It took approximately 20 min, and participants could withdraw at any time.

The article reflects data collected through self-reported questionnaires. Data on the training conditions and possibilities, attitudes, and use of healthcare were collected through a questionnaire composed by the authors of the article, and were tested (pilot study) using a representative sample including 5 Estonian elite athletes (skiers). The Emotional State Questionnaire (EST-Q2), which is validated and developed in Estonia [10], was used as a screening scale for depression and anxiety. It includes the following subscales: depression (n of items = 8; score > 11), anxiety (n = 6; >11), agoraphobia–panic (n = 5; >6), fatigue (n = 4; >6), and insomnia (n = 3; >5), reflecting symptoms of depressive and anxiety disorders according to ICD-10 and DSM-IV. This study analyzed the results of depression, anxiety, fatigue, and insomnia as these subscales reflect also general distress, not only special mental health problems. Participants reported how much various problems had troubled them during the past four weeks, using the scale: 0 = not at all; 1 = seldom; 2 = sometimes; 3 = often; 4 = all the time. It is important to point out that we cannot diagnose mental disorders with the EST-Q2; the instrument helps to identify persons with symptoms that may indicate mental health problems or reflect general emotional distress.

The software program Sigma Plot for Windows version 11.0 (GmbH Formation, Germany) was used. The whole study group was divided into subgroups according to EST-Q2: (1) low distress—none of the subscale scores (depression, anxiety, fatigue, and insomnia) surpassed the referent value; (2) moderate distress—some scores were on cutoff value and/or one surpasses it; (3) high distress—most subscale values were over the cutoff value. Results are presented as means with standard deviation (SD), or percentages. Continuous data were compared with the t-test and categorical with the χ^2^ or Fisher Exact test, as appropriate, and *p* < 0.05 was considered statistically significant. Each pairwise comparison at a significance level of *p* < 0.05 underwent a Bonferroni correction.

## 3. Results

### 3.1. Baseline Characteristics

Altogether 102 (~20%; ♀ = 58) elite athletes with average age of 24.68 ± 8.55 y (minimum 16, maximum 60) filled in the questionnaires; 90.2% of them were going in for individual sport; 30.4% of the athletes had won a place in the top six in international title competitions. Among participants were more than 20 different sport representative areas; more often they went in for track and field (22.6%), aquatics (18.6%), skiing (13.8%), and orienteering (10.8%). At the beginning of the COVID-19 outbreak, 10.8% of participants trained abroad; 70.6% prepared for international championships, including 17.6% for Olympic Games. The gender and age of different EST-Q2 groups is presented in Table 1. There were no differences in participants’ age in different research groups, but in groups (divided by the EST-Q2 results) of high distress and moderate distress, there were significantly more women compared with the low distress group (respectively *p* = 0.003 and *p* = 0.019). None of the athletes in this study had pathology associated with SARS-COV-2.

### 3.2. Mental Health Symptoms

The data of EST-Q2 show that there were nearly equal numbers of athletes who were qualified as suffering low distress (n = 33), moderate distress (n = 32), and high distress (n = 37). Most frequently the athletes had above-cutoff scores of fatigue and insomnia, followed by symptoms of depression (Figure 1). The data also show that women had more distress symptoms compared with men.

### 3.3. Training Conditions during Pandemic Situation

The most disturbing issue for athletes was the closing of training centers (57.8%), and cancellation of the competitions (50%); with athletes identifying the European Championships (36.3%) and Olympic Games (19.6%). Loss of possibilities of training (n = 43.1%) and unknown future situation (n = 17.6%) were also worrisome for study participants. However, 34.3% continued training on the same level; 43.1% tried to maintain their basic fitness level and waited for the clarification of the situation, and only five (4.9%) athletes discontinued training. The training possibilities prior and during the COVID-19 situation are presented in Figure 2. Although training opportunities differed prior to and during the pandemic, there were no differences in proportions of EST-Q2 distress groups across training opportunities. Half of the participants considered new alternative training conditions as interesting/challenging (52%), but one-fifth (21.6%) evaluated the new conditions as tiresome. Among those who considered the new situation challenging (as an opportunity to train and prepare in unique ways), were statistically less persons with depression (*p* = 0.034) compared with others. Athletes who considered the situation tiresome had more insomnia in comparison with others (59.1% vs. 31.3%; *p* = 0.032). An equal number of athletes (14) perceived new training conditions as inspiring or as demotivating.

Prior to the COVID-19 pandemic, 64.7% of athletes worked daily with a personal coach. However, during the pandemic, the opportunity for face-to-face coaching was only 18.6%. There were no statistical differences prior to and during training with a coach in different EST-Q2 groups, but during COVID-19 athletes in all groups trained with a coach significantly less than ordinarily (Figure 3). At the same time, active tutelage by digital tools emerged. The absence of a coach had a serious negative impact for the smoothness of the training process for 9.8% of athletes; for 32.4% of athletes, the absence did not affect the training results. 

Seventy-point-six percent of athletes were worried about cancellation of the competitions, but at same time the same number of participants was sure that they could be in good shape if the competitions occurred. Nearly half (41.2%) of participants planned to continue their sport careers; 27.5% planned to train even harder than ever. Unfortunately, to the question “Has the COVID pandemic influenced your thoughts about ending your athletic career?”, 27.5% responded that they thought about ending their career and 2.9% (n = 3) were certain that they would finish training conclusively. In low, moderate, and high distress groups, 15.2%, 21.9%, and 43.2% had thought about ending their career (frequency in high distress group in comparison with low, *p* = 0.022); and 0%, 3.1%, and 5.4%, respectively, were certain about it. The COVID-19 pandemic caused financial difficulties, which were directly associated with the inability to compete professionally, for 40.2% of athletes: 9.8% lost a sponsor; 11.8% had their income cut (prize money, the support from national sports federation; for 24.5% finding sponsors was more difficult than ever.

During COVID-19, athletes could not visit different healthcare specialists (Table 2) and this was unpleasant for 64.7% of participants; 31.4% of athletes confessed to not being able to get assistance for their health problems, while 33.3% said assistance was available to some extent. Athletes’ anxieties about health and emotional support sources are presented in Table 2.

## 4. Discussion

Based on the EST-Q2 results, the emotional state of our study group was worrisome—two-thirds of the athletes had symptoms of emotional distress. Female athletes had more distress than male athletes. Athletes [11] and females [12] have previously been shown to be an at-risk subpopulation for mental health problems. Additionally, it has been shown by others [9] that COVID-19 pandemic strategies are associated with poor mental health symptoms among elite athletes.

The information about mental health and wellbeing of elite athletes is limited by a paucity of high-quality, systematic studies; nevertheless, athletes are probably vulnerable to a range of mental health problems [13]. In recent years, the recognition of the mental health problems in elite athletes has improved, hopefully leading to a more holistic approach to maintain health in this population [6]. Still, in 2020 Balcombe et al. [11] claimed that elite athletes have rates of mental illness that are comparable to the prevalence in the general population. Toresdahl ja Asif [7] studied athletes’ mental health conditions during COVID-19 and reported that uncertainty and anxiety about what is going to happen had the most impact on the athletes’ mental health. Our findings were similar. The removal of social support networks and a normal training routine likely contributed to depression and anxiety.

Elite athletes train and compete in sporting mega events [14]. The organizational consequences of the quarantine/isolation include the absence of organized training and competition, lack of adequate communication between athletes and coaches, inability to move freely, lack of adequate sunlight exposure, and inappropriate training conditions [15,16]. Our results confirmed that elite athletes need working training centers and functional competition calendars to achieve healthy professional lifestyle, sustain incomes, and help them fulfill their sense of purpose. The changes in their training possibilities due to COVID-19s were not in themselves associated with mental health problems. It was the attitudes of athletes towards these changes that resulted in emotional distress. For instance, those athletes who considered the changes challenging had significantly fewer depressive symptoms compared to others.

According to Athlete 365 survey findings conducted in May 2020 [17], without a control group (more than 4000 respondents from 135 countries), 50% of athletes were struggling to keep motivated; 32% had problems with maintaining mental health, and 32% with continuing sporting careers. At same time, Estonian elite athletes had the same challenges. Nowadays usually athletes have multiprofessional support teams for facing the challenges, which consist of qualified coaches, physiotherapists, physicians, psychologists, chiropractors, masseuses, administrators, etc., but also according to Bianco [18], family, friends, and teammates. It is important for athletes’ general health, but unfortunately in the pandemic situation one-third of our athletes equally confessed not to be able to get assistance for their health problems or were not to be able get it to some extent. Mostly the athletes got emotional support from their family, but approximately 17% did not get the emotional support they needed; more lack of support was observed in moderate and high distress groups. Athletes with higher distress thought more about finishing their career. So our results confirm the idea that it is really important to provide mental health support for athletes, telehealth consultation with a sports psychologist, and encouraging maintenance of social interactions with family, friends, and teammates by phone or video chat [7,11]. Serious attention should be paid to those athletes who feel a loss of motivation, who weigh in their mind finishing their career, or who feel they do not get the emotional support they need.

Coaches have a strong influence on an athlete’s activities, performance, and the development of athletes’ personalities [19,20]. In spring 2020, most Estonian athletes could not continue everyday face-to-face collaboration. Fortunately, virtual coaching was possible such that the absence of a physical coach did not affect the athletes’ training results.

Because the pandemic caused a perceived lack of control, finding a way to take back control [21] can be a strategy to alleviate anxiety for elite competitive athletes and everyone else. For example, Jukic et al. [22] suggested that isolation is an opportunity for complete psychological and mental reset and part of athletes’ integral development. It is critical to organize sports skills trainings based on the athlete’s needs. Mental fatigue monitoring and mental training (mental self-help techniques) [23], including individual/group psychotherapy with the support of a psychologist by telecommunication should be incorporated into athletes’ daily routine. Athletes travel frequently, so probably they also will need telehealth assistance after the COVID-19 situation has eased [24]. One suggested self-help technique to promote emotional processing and reduce emotional distress is the emotional expression protocol of Written Emotional Disclosure [25], wich helps to evaluate daily routines, maintain athletes’ mental health, and support their return to competition [26].

As recently as ten years ago, athletes did not seek support for mental health problems, for reasons such as stigma, lack of understanding about mental health and its potential influence on performance, and the perception of seeking help being a sign of weakness [27,28]. In 2018 Souter et al. [29] also reported that male athletes who seek out psychological problems felt that they may be considered weak, since they were under constant performance pressure and lived their life in the public eye. Similarly, in our study, none of the male athletes sought help from psychologists. So, the same problem has been extended for ten years at least. In 2019 Purcell et al. [30] proposed a comprehensive framework for elite athlete mental health that aims at helping athletes to develop a range of self-management skills; equipping key stakeholders in elite sporting environment to better recognize and respond to athletes’ mental health concerns; and highlighting the need for skilled mental health professionals in multidisciplinary teams working with athletes. The authors are hopeful that in the future, athletes are more trusting of professionals and have the courage to ask for help, especially after experienced this novel pandemic and its impacts.

There were several limitations to this study. The authors cannot directly link the pandemic with the Estonian elite athletes’ mental health symptoms as they had no data on pre-COVID mental health conditions. Depression is one of the leading causes of disability-adjusted life years among Estonians [31], so mental health problems were common in the population prior to COVID-19. According to preliminary results of a Tallinn University study conducted from 20 April till 11 May 2020 [32] (unpublished data) Estonians (n = 1252) reported more than ordinarily poor mental health symptoms: 30% indicating depression, 27% anxiety, 46% asthenia, and 33% sleep disorders. In general, older people reported fewer symptoms than younger ones; the age group 18–24 years had more symptoms indicating depression than older groups. The larger Estonian population has experienced an increase in poor mental health symptoms during COVID-19, which is reflected in the study group. However, the frequency of reported symptoms was higher in the study group compared to the general population.

## 5. Conclusions

Mental health disorders are present globally, including within the Estonian population. The findings on Estonian elite athletes’ mental health are worrisome—two-thirds of the athletes had symptoms of emotional distress. Elite athletes need working training centers, functional competition calendars, and adequate emotional support. The impacts of the pandemic and the resulting changing environment require the sports community to make forward-looking plans to support athletes’ athletic activities and mental health needs.

## Figures and Tables

**Figure 1 ijerph-18-04317-f001:**
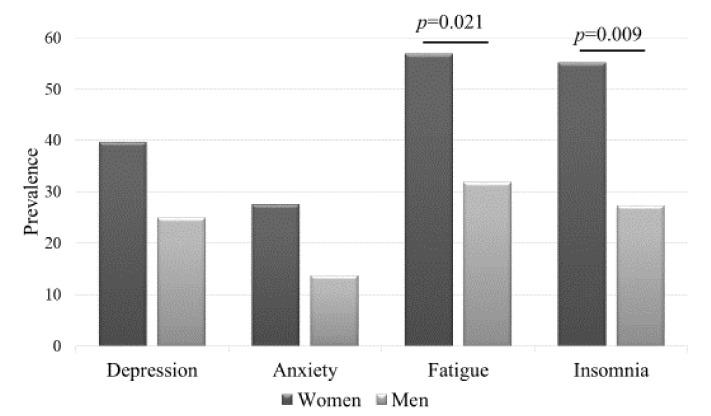
The prevalence of symptoms of depression, anxiety, fatigue, and insomnia in different gender groups.

**Figure 2 ijerph-18-04317-f002:**
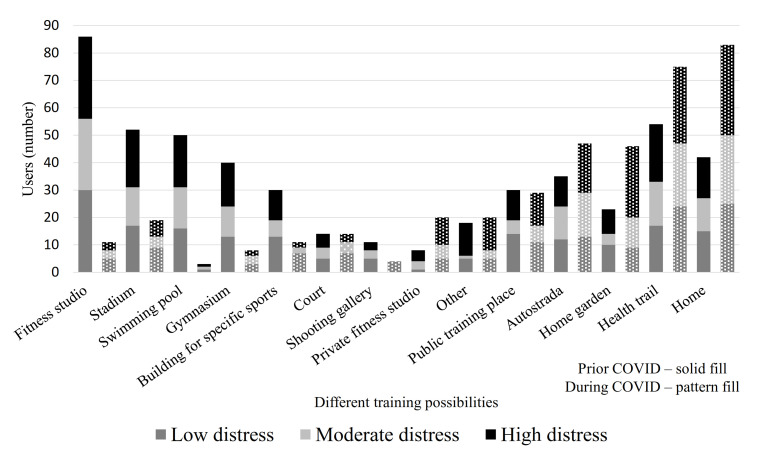
Training opportunities prior to and during the COVID-19 situation divided by results of EST-Q2.

**Figure 3 ijerph-18-04317-f003:**
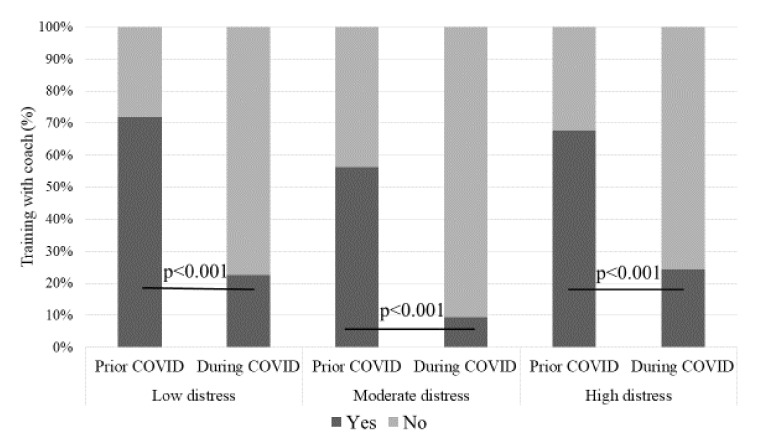
Training with a coach prior to and during the COVID-19 period in different study groups.

**Table 1 ijerph-18-04317-t001:** The age and gender in different EST-Q2 groups.

	EST-Q2			Female	Male
	Low Distress	Moderate Distress	High Distress		
n (%)	33 (32.3)	32 (31.4)	37 (36.3)	58 (56.9)	44 (43.1)
Female; n (%)	12 (36.4)	15 (46.9)	31 (83.8)	58	0
Age; mean (SD)	24.62 (8.58)	24.82 (8.84)	24.65 (8.59	24.68 (8.55)	24.78 (8.66)

**Table 2 ijerph-18-04317-t002:** COVID-19 related anxieties, lack of healthcare contacts, and sources of emotional support among athletes.

		All Study Group	Low Distress	Moderate Distress	High Distress	Women	Men
Absence of possibilities to visit (%)	physician	32.4	36.4	25	35.1	31.0	34.1
	physiotherapist	43.1	45.5	40.1	43.2	36.2	52.3
	visiting masseuse	45.1	33.3	50	51.4	48.3	40.9
	psychologist	4.9	3.0	0	10.8	8.6	0
	dentist	25.5	24.2	31.3	21.6	32.8	15.9
	chiropractic	2.0	0	0	5.4	1.7	2.3
Anxiety about their health (%)	yes	7.8	6.1	0	16.2	6.9	9.1
	to some extent	34.3	24.2	46.9	32.4	36.2	31.8
Anxiety about their family health (%)	yes	46.1	45.5	46.9	46.0	48.3	43.2
	to some extent	34.3	33.3	40.6	29.7	27.6	43.2
Emotional support from (%)	coaches	33.3	24.2	25.0	48.6	31.0	36.4
	friend	18.6	24.2	15.6	16.2	19.0	18.2
	training member	2.9	3.0	0	5.4	3.4	2.3
	family members	52	63.6	50.5	43.2	50.0	54.5
	mentor	3.9	0	3.1	8.1	3.4	4.5
	psychologist	8.8	3.0	15.6	8.1	10.3	6.8
	no support	16.7	6.1	21.9	21.6	19.0	13.6

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
