# Peer review of "Impact of the COVID-19 Pandemic on Estonian Elite Athletes: Survey on Mental Health Characteristics, Training Conditions, Competition Possibilities, and Perception of Supportiveness"

_ijerph, 2021, doi:10.3390/ijerph18084317_

Round 1

Reviewer 1 Report

Authors presented an interesting study on COVID impacts in a unique population of elite athletes. Reviewer has provided rewrites for most sections of the article including line numbers/page numbers. Please see additional comments and questions to improve article in the attached Word document. A major concern is lack of information about the whether the survey tool was previously validated or if results were compared to a validated method.

Author Response

Dear reviewer,

here are our comments.

Response to Reviewer 1 Comments

Point 1: Please see additional comments and questions to improve article in the attached Word document.

Response 1: Thank you for the comment, but unfortunately we could not find any Word document from the electronic system. Hopefully these comments are added to the text.

Point 2: A major concern is lack of information about the whether the survey tool was previously validated or if results were compared to a validated method.

Response 2: One part of the questionnaire was created to get background information about Estonia elite athletes training conditions during COVID situations. It was done in purpose to use previously validated The Emotional State Questionnaire (EST-Q2) (Aluoja, A.; Shlik, J.; Vasar, V.; Luuk, K.; Leinsalu, M. Development and psychometric properties of the Emotional State Questionnaire, a self-report questionnaire for depression and anxiety. Nord J Psychiatry 1999, 53, 443–449) to get information about their emotional wellbeing. The survey tool was tested (pilot study) using a representative sample including 5 Estonian elite athletes (skiers).

Best wishes,

Anna-Liisa

Reviewer 2 Report

The current pandemic situation has affected the mental health of the entire population. Some groups, such as elite athletes, were particularly affected by restrictions by government organizations in place to control the spread of the disease. This fact has generated an increase in the incidence of mental pathologies associated with stress, anxiety and depression, which must be evaluated in depth.

According to authors (page 7, line 235), one of the main limitations of the study is not reflecting the starting situation of the individuals analyzed. Another issue to consider is the existence of other variables that could be affecting mental health, apart from sports activity. Do any of the athletes perform other work activities that could have been affected by the pandemic? In addition, there is no mention of whether any of the athletes suffered any pathology associated with SARS-COV-2 that would have affected their physical performance. This greatly limits the interpretation of the results at the statistical level, and prevents a good association between the variables analyzed.

On the other hand, the conclusions of the study (page 7, line 246) generically analyze the current situation due to the pandemic. Direct and clear conclusions derived from the study results are not reflected. It would be convenient for the authors to rewrite a section in which the conclusions of the study are drawn. And following this line, the summary of the article (page 1, line 23) should also reflect these conclusions.

As minor observations:

  • In Table 1 (page 3, line 101) it would be convenient to indicate that age is expressed as a mean, not an average.
  • On page 4, line 137, the authors report that the absence of the coach affected the quality of training. What data was taken into account to assess the quality of the training of the individuals?
  • On page 5, line 144, the authors reflect that almost 30% of the sample are thinking about ending their sports career or that they have already made the decision. On the other hand, the percentage of individuals who had thought about finishing it was determined depending on the degree of distress. How do the authors directly associate the decision to end a sports career with the pandemic situation? Would some of these athletes have made this decision without COVID-19?
  • On page 5, line 153, a % may be missing towards the end of the line.
  • On page 5, table 2, the last two columns (women and men) are not wide enough. This is likely due to formatting issues.
  • On page 6, line 201, the authors reflect that the athletes with the greatest distress thought more about finishing their sports career and complained about the absence of masseuse and psychologist, and that these results reinforce the idea of ​​the need for health support mental. It is necessary for the authors to clarify this point, since no specific information has been given on the athletes who will leave their career due to the pandemic or for reasons unrelated to it.

Best regards.

Author Response

Dear reviewer,

here are our comments.

Response to Reviewer 2 Comments

Point 1: Another issue to consider is the existence of other variables that could be affecting mental health, apart from sports activity. Do any of the athletes perform other work activities that could have been affected by the pandemic?

Response 1: The study group consisted of elite athletes whose life and income is mostly associated with sport activity. Thank you for this comment. This is one limitations of our study – we did not ask their labour and study situations. We know that “COVID-19 pandemic situation caused financial difficulties for 40.2% of athletes: 9.8% lost the sponsor; 11.8% had their income cut; for 24.5% finding sponsors was more difficult than ever“.

Point 2: On the other hand, the conclusions of the study (page 7, line 246) generically analyze the current situation due to the pandemic. Direct and clear conclusions derived from the study results are not reflected. It would be convenient for the authors to rewrite a section in which the conclusions of the study are drawn. And following this line, the summary of the article (page 1, line 23) should also reflect these conclusions.

Response 2: Conclusion: “Mental health disorders are present globally, including within the Estonian population. The findings on Estonian elite athletes` mental health are worrisome – two-thirds of the athletes had symptoms of emotional distress. Elite athletes need working training centers, functional competition calendars and adequate emotional support. The impacts of the pandemic and the resulting changing environment requires the sports community to make forward-looking plans to support athletes` athletic activities and mental health needs.“

In abstract by the reason of limit of words: “The Estonian sport community needs to adapt to life in a pandemic environment and help athletes to maintain training and competition activities and in turn, their mental health.”

In addition, there is no mention of whether any of the athletes suffered any pathology associated with SARS-COV-2 that would have affected their physical performance.

None of the athletes in this study had pathology associated with SARS-COV-2. This information is now added also to the manuscript (page 3).

Point 3: In Table 1 (page 3, line 101) it would be convenient to indicate that age is expressed as a mean, not an average.

Response 3: Thank you, this correction is done.

Point 4: On page 4, line 137, the authors report that the absence of the coach affected the quality of training. What data was taken into account to assess the quality of the training of the individuals?

Response 4: Thank you for this comment. Unfortunately, we did not assess quality of trainings, we based only on athletes self-reports. 

Point 5: On page 5, line 144, the authors reflect that almost 30% of the sample are thinking about ending their sports career or that they have already made the decision. On the other hand, the percentage of individuals who had thought about finishing it was determined depending on the degree of distress. How do the authors directly associate the decision to end a sports career with the pandemic situation? Would some of these athletes have made this decision without COVID-19?

Response 5: The question was raised as “Is the COVID pandemic has influenced your thoughts about ending your athletes´ career?” So we can associate the decision to end a sport career with the pandemic situation.

Point 6: On page 5, line 153, a % may be missing towards the end of the line.

Response 6: Thank you, this correction is done.

Point 7: On page 5, table 2, the last two columns (women and men) are not wide enough. This is likely due to formatting issues.

Response 7: Thank you, this correction is done.

Point 8: On page 6, line 201, the authors reflect that the athletes with the greatest distress thought more about finishing their sports career and complained about the absence of masseuse and psychologist, and that these results reinforce the idea of ​​the need for health support mental. It is necessary for the authors to clarify this point, since no specific information has been given on the athletes who will leave their career due to the pandemic or for reasons unrelated to it.

Response 8: Thank you for this comment. We removed the second part of the sentence “Athletes with higher distress thought more about finishing the career and complained about the absence of masseuse and psychologist” (difference was not statistically significant).

Best wishes,

Anna-Liisa

Reviewer 3 Report

The current study explored the possible relationships between mental health characteristics, training conditions, competition possibilities, and received support among Estonian elite athletes. While the topic of the current study is timely and relevant, there are several issues to be addressed. Firstly, it is not sure whether the mental health characteristics (or conditions) are truly affected by the Covid-19 situation. The authors may need to provide some reference statistics (from the previous studies) about the typical mean scores of the ESQ indexes, etc. Secondly, it was not certain whether the authors tried to argue whether the Covid-related situations (e.g., support, competition possibilities) lead to mental health situations or the other way around. From a logical perspective, the perceived COVID-19 related situations would influence athletes' mental health situations. However, some of the analyses were conducted the other way around. Thirdly, the authors need to provide more detailed information about the measures and analyses. For example, how many questions were included in the EST-Q2, how many questions were asked per each dimension of the EST-Q2? how did you come up with three EST-Q2 groups? what do you mean by "above-cutoff scores (of fatigue and insomnia)? There are so many things that need to be explained in the method and result sections. In addition, you may need to revise Table 1 to include the gender differences (e.g., mean EST-Q2 scores of men and women). Also, you need to provide a new table to visualize what you reported in lines 141-151. Lastly, the manuscript should be thoroughly proofread by a native speaker. There are so many grammatical errors and editorial mistakes (e.g., the use of contractions; missing articles). Also, not many people would know what 'the first six first grades' means. 

Author Response

Dear reviewer,

here are our comments.

Response to Reviewer 3 Comments

Point 1: Firstly, it is not sure whether the mental health characteristics (or conditions) are truly affected by the Covid-19 situation. The authors may need to provide some reference statistics (from the previous studies) about the typical mean scores of the ESQ indexes, etc.

Response 1: According to preliminary results of Tallinn University conducted in May 2020 [unpublished data; same time as our study] among Estonians reported more than ordinarily mental health symptoms: 30% indicating depression, 27% anxiety, 46% asthenia and 33% sleep disorders. According to our study, all these parameters were even higher in female elite athletes. Unfortunately, we cannot directly link the pandemic with the Estonian elite athletes` mental health symptoms as we had no data on pre-COVID mental health conditions. This information is presented also in manuscript as limitation.

Point 2: Secondly, it was not certain whether the authors tried to argue whether the Covid-related situations (e.g., support, competition possibilities) lead to mental health situations or the other way around. From a logical perspective, the perceived COVID-19 related situations would influence athletes' mental health situations. However, some of the analyses were conducted the other way around.

Response 2: Thank you for the comment! Unfortunately, we cannot find the data which shows the influence of emotional wellbeing to COVID-19 related situation. Could you, please, specify!

Point 3: Thirdly, the authors need to provide more detailed information about the measures and analyses. For example, how many questions were included in the EST-Q2, how many questions were asked per each dimension of the EST-Q2?

How did you come up with three EST-Q2 groups? What do you mean by "above-cutoff scores (of fatigue and insomnia)?

Response 3: Thank you for the comment, number of items and scores are added: “The Emotional State Questionnaire (EST-Q2) was developed in Estonia [10] as a screening scale for depression and anxiety. It includes the following subscales: depression (n of items =8; score >11), anxiety (n=6; >11), agoraphobia-panic (n=5; >6), fatigue (n=4; >6), and insomnia (n=3; >5), reflecting symptoms of depressive and anxiety disorders according to ICD-10 and DSM-IV.”

Point 4: In addition, you may need to revise Table 1 to include the gender differences (e.g., mean EST-Q2 scores of men and women).

Response 4: Thank you for the comment, but the gender differences in EST-Q2 are presented in Figure 1.

Point 5: Also, you need to provide a new table to visualize what you reported in lines 141-151.

Response 5: Thank you for the comment, but in lines 141-151 is text 70.6% of athletes were worried about cancellation of the competitions, but at same time the same amount of participants was sure that they could be in good shape if the competitions eventuate. Nearly half (41.2%) of participants did not think about finishing the sport careers; 27.5% planned to train even harder than ever. Unfortunately, 27.5% thought about finishing the career and 2.9% (n=3) were certain that they would finish training conclusively. In low, moderate and high distress group there were 15.2%, 21.9% and 43.2% of those, who had thought about finishing the career (frequency in high distress group in comparison with low; p=0.022); and 0%, 3.1% and 5.4%, respectively, who were certain about it. COVID-19 pandemic situation caused financial difficulties for 40.2% of athletes: 9.8% lost the sponsor; 11.8% had their income cut; for 24.5% finding sponsors was more difficult than ever.”, which we think is unnecessary to duplicate in text and in table.

Point 6: Lastly, the manuscript should be thoroughly proofread by a native speaker. There are so many grammatical errors and editorial mistakes (e.g., the use of contractions; missing articles). Also, not many people would know what 'the first six first grades' means. 

Response 6: Thank you, these corrections are done; 'the first six first grades' is changed to „30.4% of the athletes had won a place in the top six in international title competitions“.

Point 7: Did the Authors consider to test possible differences between individual and team sport athletes?

Response 7: Yes, but unfortunately there were only 10 team sport athletes, so the number was too small for statistical analysis.

Point 8: Please ameliorate the quality of the Figures (in terms of pixels definition).

Response 8: Thank you for the comment. We really hope that the quality in original Figures is proper for the Journal.

Point 9: Table 1. The second row should be: “Female; n (%)”

Response 9: Thank you, this correction is done.

Point 10: Conclusions: please include possible practical applications/suggestions derived from the findings of the study.

Response 10: Mental health disorders are present globally, including within the Estonian population. The findings on Estonian elite athletes` mental health are worrisome – two-thirds of the athletes had symptoms of emotional distress. Elite athletes need working training centers, functional competition calendars and adequate emotional support. The impacts of the pandemic and the resulting changing environment requires the sports community to make forward-looking plans to support athletes` athletic activities and mental health needs.

Best wishes,

Anna-Liisa

Reviewer 4 Report

The aim of the present study was to assess mental health characteristics, training conditions, competition possibilities, and received support in Estonian elite athletes during COVID-19-related situation. The study is well conducted and provides a comprehensive picture of both mental and behavioural status of elite athletes in Estonia during COVID. The findings of the study are relevant within the context of mental health-related problems in athletes. As stated within the Discussion, mental health-symptoms are widely present in Estonians, and this could also be exacerbated for elite athletes. The manuscript is well composed. I have only a few minor comments that I hope will be useful to improve the scientific quality of the manuscript. Finally, I strongly suggest to revise the language of the manuscript.

  • Did the Authors consider to test possible differences between individual and team sport athletes?
  • Please ameliorate the quality of the Figures (in terms of pixels definition).
  • Table 1. The second row should be: “Female; n (%)”
  • Conclusions: please include possible practical applications/suggestions derived from the findings of the study.

Author Response

Dear reviewer,

here are our comments.

Response to Reviewer 4 Comments

Point 1: I strongly suggest to revise the language of the manuscript.

Response 1: Thank you for the comment, the corrections are done.

Point 2: Table 1. The second row should be: “Female; n (%)”

Response 2: Thank you, this correction is done.

Point 3: Conclusions: please include possible practical applications/suggestions derived from the findings of the study.

Response 3: Mental health disorders are present globally, including within the Estonian population. The findings on Estonian elite athletes` mental health are worrisome – two-thirds of the athletes had symptoms of emotional distress. Elite athletes need working training centers, functional competition calendars and adequate emotional support. The impacts of the pandemic and the resulting changing environment requires the sports community to make forward-looking plans to support athletes` athletic activities and mental health needs.

Best wishes,

Anna-Liisa

Round 2

Reviewer 1 Report

Thank you for the response and for notifying that you did not see the attachment with my comments. I have reattached them for your convenience. I do not need to review again.

Author Response

Dear reviewer!

Please excuse our mistake, we could not find the file previously.

Best wishes,

Anna

Reviewer 2 Report

Regarding response 1, the authors identified another study limitation. It was determined that 40.2% of the athletes presented financial difficulties. However, it is not directly associated with the inability to compete professionally. No information has been recorded on the work activity of the athletes, regardless of their sports career. This information does not appear in the final paragraph of the discussion (page 7, line 232).

Analyzing response 4, if no tool was used to determine the quality of the training, the authors cannot affirm that the absence of the coach had a significant negative impact on the quality of the training. If the recording of the quality of the training has only been based on a subjective interpretation of the athletes, this must be properly explained.

In response 5 the authors explain that the information record was obtained from the question "Is the COVID pandemic has influenced your thoughts about ending your athletes' career?". This should be clarified in the article to reflect that the decision to end a sports career is associated with the difficulties caused by the pandemic and not by other factors.

Best regards

Author Response

Thank you for the excellent commentes. All corrections are done to the manuscript.

Best wishes,

Anna

Reviewer 3 Report

The revised manuscript reads better. However, the authors need to discuss more the managerial/practical strategies to cope with the Covid-19 situation while integrating a few critical pieces of literature, such as:

Readon et al. (2020). Mental health management of elite athletes during COVID-19: a narrative review and recommendations. 

Davis et al. (2020). Written Emotional Disclosure Can Promote Athletes’ Mental Health and Performance Readiness During the COVID-19 Pandemic. 

Schinke et al. (2020). Sport psychology services to high performance athletes during COVID-19

Leguizamo et al. (2021). Personality, Coping Strategies, and Mental Health in High-Performance Athletes During Confinement Derived From the COVID-19 Pandemic.

You may also use these pieces to strengthen your argument in the introduction section. 

Author Response

Dear reviewer,

thank you for the suggestion. We did some corrections to the manuscript.

Best wishes,

Anna
